# Early Dysfunction of Substantia Nigra Dopamine Neurons in the ParkinQ311X Mouse

**DOI:** 10.3390/biomedicines9050514

**Published:** 2021-05-05

**Authors:** Maria Regoni, Letizia Zanetti, Stefano Comai, Daniela Mercatelli, Salvatore Novello, Federica Albanese, Laura Croci, Gian Giacomo Consalez, Andrea Ciammola, Flavia Valtorta, Michele Morari, Jenny Sassone

**Affiliations:** 1Division of Neuroscience, San Raffaele Scientific Institute, 20132 Milan, Italy; regoni.maria@hsr.it (M.R.); zanetti.letizia@hsr.it (L.Z.); stefano.comai@unipd.it (S.C.); croci.laura@hsr.it (L.C.); consalez.giangiacomo@hsr.it (G.G.C.); valtorta.flavia@hsr.it (F.V.); 2Vita-Salute San Raffaele University, 20132 Milan, Italy; 3Department of Pharmaceutical and Pharmacological Sciences, University of Padua, 35122 Padua, Italy; 4Department of Biochemical Sciences, University of Padua, 35122 Padua, Italy; 5Department of Neuroscience and Rehabilitation, Section of Pharmacology, University of Ferrara, 44121 Ferrara, Italy; mrcdnl@unife.it (D.M.); salvatore.novello@epfl.ch (S.N.); lbnfrc@unife.it (F.A.); mri@unife.it (M.M.); 6Department of Neurology and Laboratory of Neuroscience, Istituto Auxologico Italiano IRCCS, 20149 Milan, Italy; andrea.ciammola@auxologico.it

**Keywords:** Parkinson’s disease, dopaminergic neurons, parkinQ311X mouse, early dysfunction, mitochondria, cytoplasmic vacuolization, firing activity

## Abstract

Mutations in the *PARK2* gene encoding the protein parkin cause autosomal recessive juvenile parkinsonism (ARJP), a neurodegenerative disease characterized by early dysfunction and loss of dopamine (DA) neurons in the substantia nigra pars compacta (SNc). No therapy is currently available to prevent or slow down the neurodegeneration in ARJP patients. Preclinical models are key to clarifying the early events that lead to neurodegeneration and reveal the potential of novel neuroprotective strategies. ParkinQ311X is a transgenic mouse model expressing in DA neurons a mutant parkin variant found in ARJP patients. This model was previously reported to show the neuropathological hallmark of the disease, i.e., the progressive loss of DA neurons. However, the early dysfunctions that precede neurodegeneration have never been investigated. Here, we analyzed SNc DA neurons in parkinQ311X mice and found early features of mitochondrial dysfunction, extensive cytoplasmic vacuolization, and dysregulation of spontaneous in vivo firing activity. These data suggest that the parkinQ311X mouse recapitulates key features of ARJP and provides a useful tool for studying the neurodegenerative mechanisms underlying the human disease and for screening potential neuroprotective drugs.

## 1. Introduction

Mutations in the *PARK2* gene encoding the protein parkin are the most frequent cause of autosomal recessive juvenile parkinsonism (ARJP) (OMIM# 600116). The development and characterization of preclinical models that mimic the human disease are essential for identifying the cellular and molecular mechanisms underpinning the disease, target validation, and in vivo screening of novel pharmacological and non-pharmacological therapeutic approaches. Many patients display exon deletions that result in a complete loss of the parkin protein [1]; accordingly, parkin knockout mice have been generated. Parkin knockout mice display mild nigrostriatal, cognitive, and noradrenergic dysfunctions [2,3,4,5] but do not show the neuropathological hallmark of Parkinson’s disease (PD), i.e., the loss of nigrostriatal DA neurons [2,3,4,5]. Therefore, they cannot be used to test the efficacy of disease-modifying drugs designed to halt neuronal loss and increase neuron survival. Since some *PARK2* patients bear point mutations that lead to the expression of mutant parkin forms [1], the study of transgenic mice overexpressing pathological parkin variants has been proposed as an attractive alternative to parkin deficiency models for understanding neurodegeneration and developing novel therapeutics [6,7,8].

ParkinQ311X is a bacterial artificial chromosome (BAC) transgenic mouse expressing a C-terminal truncated human mutant parkin found in ARJP kindred under the DA transporter (DAT) promoter [7]. This parkinQ311X model exhibits SNc DA neuron degeneration at 6 [9] and 16 [7] months of age. Identifying early pathological events preceding neurodegeneration in this model can help identify potential targets for neuroprotection. With this study, we analyzed the time course of nigrostriatal neurodegeneration and identified a time window with no overt neurodegeneration. At this time-point, we analyzed neuronal activity and mitochondrial morphology in the search for an early dysfunction of DA neurons. In detail, we looked at cellular and biochemical dysfunctions previously found in DA neurons derived from human-induced pluripotent stem cells (iDA) of patients bearing *PARK2* mutations, such as mitochondrial defects [10,11], cytoplasmic vacuolization [12], and abnormal bursting activity [13].

## 2. Experimental Section

### 2.1. Animals

C57B1/6N ParkinQ311X mice were previously generated and characterized [7]. This mouse model has normal wild-type (WT) parkin alleles on both chromosomes in addition to the exogenous human parkinQ311X variant [14], which is expressed by a BAC and driven by the DAT promoter. The BAC transgene is integrated in two copies in tandem. The expression level of the mutant human parkinQ311X is 42% of the murine endogenous parkin [7]. This mouse phenotype results from a dominantly inherited toxic gain of function [7]. Control mice were gender-matched littermates. In compliance with international guidelines for animal research, all experiments were planned and conducted so as to minimize the number of animals sacrificed. Since *PARK2*-related ARJP is not sex-linked and the prevalence and disease symptoms do not differ between males and females, according to the principle of the 3Rs (replacement, reduction, refinement), both male and female animals (in equal ratio) were used to minimize the number of animals used in this study.

Mice were maintained and bred at the animal facility of the San Raffaele Scientific Institute under constant temperature (22 ± 1 °C) and humidity (50%) conditions, with a 12 h light/dark cycle (light on at 7 AM and light off at 7 PM) and were provided with food and water ad libitum. All experiments involving animals were conducted in compliance with institutional guidelines and international laws (EU Directive 2010/63/EU EEC Council Directive 86/609, OJL 358, 1, December 12, 1987, NIH Guide for the Care and Use of Laboratory Animals, U.S. National Research Council, 1996) and were approved by the Institutional Animal Care and Use Committee (IACUC) of the San Raffaele Scientific Institute and the Italian Ministry of Health (IACUC 670/2017-PR and 53/2021-PR).

### 2.2. Mouse Genotyping

DNA was extracted from mouse tails at 7 days of age. Briefly, the tails were incubated overnight at 56 °C in a 500 μL Digestion Buffer (50 mM Tris-HCl pH 8, 100 mM EDTA pH 8, 100 mM NaCl, 1% SDS) containing 10 mg/mL Proteinase K (Sigma-Aldrich, St. Louis, MO, USA) for the enzymatic digestion of proteins and non-nucleic acid cellular components. A mixture of phenol:chloroform:isoamyl alcohol (25:24:1) (Sigma-Aldrich, St. Louis, MO, USA) was then added to promote the partitioning of lipids and cellular debris into the organic phase, leaving the isolated DNA in the upper aqueous phase. Following centrifugation at 13,000× *g*, the aqueous phase containing DNA was transferred to a clean tube and precipitated with isopropanol. After centrifugation, the DNA pellet was washed with a 70% ETOH solution and centrifuged again at 13,000× *g*. Finally, DNA was resuspended in Tris-EDTA Buffer 1X (TE, 10 mM Tris HCl pH 7.4, 1 mM EDTA pH 8) and quantified using Nanodrop (ThermoFisher, Waltham, MA, USA). Genotyping was performed by PCR with the following primers: Internal Control IL2 gene FOR (CTAGGCCACAGAATTGAAAGATCT), Internal Control IL2 gene REV (GTAGGTGGAAATTCTAGCATCATCC), ParkinQ311X FOR (ATGGACTACAAAGACGATGACGACAAG), ParkinQ311X REV (ATTCTGCACAGTCCAGTCATTCCTC), in order to detect the internal control IL2 gene (product, 324 bp), and the mutant Parkin transgene (product, 200 bp). We used the following PCR conditions: 3 min at 95 °C, 35 cycles (30 s at 95 °C, 30 s at 59 °C, 1 min at 72 °C), followed by 10 min at 72 °C.

### 2.3. Dissection of Substantia Nigra

Mice at 1 month of age were anesthetized by an intraperitoneal (i.p.) injection of ketamine/xylazine (100 and 10 mg/kg, respectively, Sigma-Aldrich, St. Louis, MO, USA) and sacrificed by cervical dislocation. Their brains were washed in ice cold phosphate buffered saline (PBS) for 2–3 min and transferred to the platform of a tissue chopper (McIlwain™ Tissue Chopper, Mickle Laboratory Engineering, Guildford, UK). The tissue was positioned perpendicularly to the chopper blade, and coronal slices were cut (300 μm thick) in less than 1 min. The brain slices were transferred to PBS 1X. The substantia nigra was isolated under a dissecting microscope and immediately frozen in liquid nitrogen.

### 2.4. Stereological Cell Count in SNc

Stereological counting was performed according to published methods [9,15]. The mice were deeply anesthetized by an i.p. injection of ketamine/xylazine (100 and 10 mg/kg, respectively, Sigma-Aldrich, St. Louis, MO, USA) and transcardially perfused with an ice-cold 0.9% NaCl solution, followed by 4% paraformaldehyde (PFA) in PBS (0.1 M, pH 7.4, Sigma-Aldrich). The brains were postfixed in 4% PFA/PBS overnight at 4 °C, transferred to a 30% sucrose solution in PBS for cryoprotection for 24–48 h, and then frozen in isopentane and stored at −80 °C. Free-floating 50-μm-thick sections of SNc (AP from −2.80 to −3.80 from bregma) [16] were cut on a freezing cryostat. The sections were washed in PBS and incubated in 3% hydrogen peroxide/PBS for 10 min to quench the endogenous peroxidase activity. After several washes, the sections were incubated for 30 min with blocking solution (PBS + bovine serum albumin [BSA] 1:50 + Triton X-100 0.3%) and then incubated overnight at room temperature with an anti-tyrosine hydrolase [TH] antibody (ab112, 1:750 in 1% BSA PBS with Tween^®^ 20 [PBST], Abcam, Cambridge, UK). The sections were then rinsed, incubated for 1 h with an anti-rabbit horseradish peroxidase [HRP]-conjugated secondary antibody (ab6721, 1:2000 in BSA 1% PBST, Abcam, Cambridge, UK), and revealed with a DAB (3,3’-diaminobenzidine) substrate kit (ab64238, Abcam, Cambridge, UK). The sections were mounted on gelatin-coated slides and stained with cresyl violet. A stereological analysis was performed by counting TH-positive neurons (phenotypic marker) and cresyl violet-stained cells (structural marker) in the SNc. Neural cell counting was performed on five serial slices (magnified at 40×), which were cut 50 μm thick and 200 μm apart by applying an unbiased stereological sampling method based on an optical fractionator stereological probe [15,17,18]. We used a Leica DM600B motorized microscope (Leica, Wetzlar, Germany) equipped with the Stereo Investigator software (MBF Europe, Delft, The Netherlands). The number of neurons is expressed as a raw value derived from a software calculation that estimates the total number of neurons within a systematic, randomly sampled set of unbiased virtual counting spaces covering the entire region of interest with a uniform distance between unbiased virtual counting in spaces in three directions (X, Y, Z). Image acquisition and quantification were performed by investigators blinded to the experimental condition.

### 2.5. Immunofluorescence

Mice were anesthetized by an i.p. injection of ketamine/xylazine (100 and 10 mg/kg, respectively, Sigma-Aldrich, St. Louis, MO, USA) and perfused with an ice-cold 0.9% NaCl solution, followed by 4% PFA in PBS (0.1 M, pH 7.4, Sigma-Aldrich, St. Louis, MO, USA). The brains were postfixed in 4% PFA overnight at 4 °C, transferred to a 30% sucrose solution in PBS, and then frozen in isopentane and stored at −80 °C. SNc sections with a thickness of 30 μm were prepared on a freezing microtome, transferred to polylysine-coated glass slides, and dried at room temperature. The sections were then rehydrated, washed, and incubated in a 10 mM sodium citrate buffer (pH 6) at 80 °C for 30 min (antigen retrieval), then kept for 20 min on ice. The slices were incubated in a blocking solution, 5% normal goat serum (Sigma-Aldrich, St. Louis, MO, USA) with 0.1% Triton X-100 in PBS, incubated for 48 h with a 1:500 anti-TH (Mab318, Millipore, Burlington, MA, USA) primary antibody. After three washes with PBS, the tissues were incubated for 2 h at room temperature in the dark with a secondary antibody (1:300 goat anti-mouse IgG Alexa 546, Invitrogen, Carlsbad, CA, USA). After three washes in PBS, cell nuclei were counterstained with a 4′,6-diamidino-2-phenylindole solution. The slides were mounted using a Dako fluorescence mounting medium (Dako, Glostrup, Denmark). Images were acquired using a Leica TCS SP2 confocal microscope (Leica) and analyzed using the ImageJ software. Image acquisition and quantification were performed by investigators blinded to the experimental condition.

### 2.6. Western Blotting

Western blotting was conducted as described elsewhere [9]. The primary antibodies were: anti-OPA1 (BD, Transduction Laboratories, Franklin Lake, NJ; USA, #612607 1:2500), anti-SDHA (Thermo Scientific, Rockford, IL, USA, #459200 1:10.000), anti-VDAC (Abcam, Cambridge, UK, #ab147 1:1000), and anti-GAPDH (Santa Cruz, Dallas, TX, USA, cod. sc-25778 1:1000).

### 2.7. Quantification of Vacuoles in DA Neurons

SNc sections (30 μm thick) were prepared on a freezing microtome and stained with an anti-TH antibody as described above. Epifluorescence images were acquired using a Zeiss Axio Vision microscope equipped with an AxioCam MRm camera (Carl Zeiss, Oberkochen, Germany). The vacuolated TH+ cells were identified; the vacuoles were marked using the overlay free-hand selection tool, and their areas calculated using the ROI manager tool in ImageJ. Finally, the number of vacuoles in each cell was counted.

### 2.8. In Vivo Single-Unit Extracellular Recording of SNc DA Neurons

The in vivo single-unit extracellular recording of SNc dopaminergic neurons was performed between 10 AM and 4 PM following validated procedures [19]. Mice were anesthetized with an i.p. injection of chloral hydrate (400 mg/kg, 0.6% solution) and placed in a stereotaxic apparatus with the skull positioned horizontally. Anesthesia was confirmed by the absence of nociceptive reflex reaction to a tail or a paw pinch. During the experiments, anesthesia was maintained by periodic i.p. injections (0.05 mL) of supplemental doses of chloral hydrate (100 mg/kg, i.p.). To expose the brain sutures, an incision was made in the scalp, and the surface was cleaned with hydrogen peroxide. A burr hole was drilled through the skull over the SNc. The stereotaxic brain coordinates of the SNc were chosen according to Paxinos and Franklin’s mouse brain atlas (2001) [16]: 0.08–2.0 mm posterior to the interaural line, 0.7–1.7 mm lateral to midline, and 3.5–5.0 mm in depth. The spontaneous electrical activity of DA SNc cells was recorded using single-barreled glass micropipettes (1.5 mm O.D. × 0.86 mm I.D., 100 mm L, Harvard Apparatus, Holliston, MA, USA) pulled on a Narashige (Tokyo, Japan) PE-2 pipette puller filled with 2% pontamine sky blue (PSB) solution in sodium acetate 0.5 M and with a tip reduced to 1–3 μm in diameter to reach an impedance from 2 to 6 MΩ. The electrode was inserted into the SNc with a manual hydraulic micropositioner (model 650, David Kopf Instruments, Tujunga, California). The firing activity of single SNc DA cells was recorded as discriminated action potentials from a filtered electrode signal amplified by a Bak Electronics Model RPI amplifier system (Umatilla, FL, USA), converted into a digital signal by an interface system (CED 1401, Cambridge Electronic Design [CED], Cambridge, UK) and analyzed off-line using the Spike2 software, version 8.21 (CED). SNc DA neurons were identified based on well-established electrophysiological properties: a wide action potential (>2.5 ms), biphasic or triphasic waveform, and slow firing rate (0.5–10 Hz). SNc DA burst-firing activity was analyzed using a script for the Spike 2 software according to previously described criteria [19,20]. A burst was defined as a train of at least two spikes with an initial interspike interval (ISI) ≤80 ms and a maximum ISI of 160 ms, within a regular low-frequency firing pattern and decreased amplitude from the first to the last spike within the burst [19,20].

### 2.9. Transmission Electron Microscopy (TEM)

WT mice (*n* = 3) and parkinQ311X mice (*n* = 4) at 1 month of age were anesthetized with ketamine/xylazine (100 and 10 mg/kg, respectively, Sigma-Aldrich, St. Louis, MO, USA) and transcardially perfused (4% paraformaldehyde and 1% glutaraldehyde in 0.1M cacodylate buffer). Brain coronal slices were cut to a 300 μm thickness using a tissue chopper. SNc tissues were isolated under a dissecting microscope. The samples were fixed for 24 h at 4 °C (4% PFA and 2.5% glutaraldehyde in 125 mM cacodylate buffer), postfixed (2% OsO4 in 125 mM cacodylate buffer), washed, and embedded in Epon. Conventional thin sections were collected on uncoated grids and stained with uranyl and lead citrate. The grids were examined under a Talos L 120C electron microscope (Thermo Fisher Scientific, Waltham, MA, USA) at 120 kv and a magnification of 22,000×. From WT mice we analyzed 74 photos, while for parkinQ311X 84 photos were analyzed. Each mitochondrion in the images was analyzed to detect ultrastructure disruption. In each image we quantified the number of mitochondria with normal morphology, mitochondria showing loss of outer membrane shape or form, and mitochondria showing a vesicular structure and swollen mitochondria. A total of *n* = 246 mitochondria were analyzed for WT and *n* = 320 mitochondria for parkinQ311X mice.

### 2.10. Data Presentation and Statistical Analysis

Data are presented as the mean ± standard error of the mean (SEM). The statistical analysis was implemented in GraphPad Prism 6 (GraphPad, San Diego, CA, USA). The Kolmogorov–Smirnov test and Bartlett’s test were used to assess variance across groups. A one-way ANOVA followed by Sidak’s multiple comparisons test was used to compare the number of DA neurons between the WT and the Q311X mice. The Mann–Whitney test was used to compare the number and area of vacuoles. A two-tailed unpaired Student’s *t* test was used to compare densitometer data from the Western blotting and firing activity between the 1-month old WT and Q311X mice. The distribution of mitochondria morphology was analyzed using the chi-square test.

## 3. Results

### 3.1. DA Neuron Loss in the SNc of ParkinQ311X Mice

To define the time course of nigral neurodegeneration, we counted the number of DA neurons in the SNc of parkinQ311X mice at 1, 6, and 12 months of age. At 1 month of age, the number of SNc DA neurons was similar in transgenic mice and their WT littermates, whereas at 6 and 12 months of age, parkinQ311X mice showed a reduction of 23% and 25% in DA neurons, respectively (Figure 1a shows representative immunofluorescence images with TH labeling in the SNc of WT and parkinQ311X mice at 1, 6, and 12 months of age; Figure 1b shows the results of the stereological count and one-way ANOVA, followed by Sidak’s multiple comparisons test F = 5.998). This was accompanied by a loss of dopaminergic dendrites in the pars reticulata of the substantia nigra (Figure 1a). These results demonstrate that the parkinQ311X transgenic strain recapitulates the early degeneration of DA neurons and a slow disease progression of nigrostriatal dysfunction typical of ARJP patients [21,22,23]. The data also indicate that because neurodegeneration is not yet evident at 1 month of age, the investigation of early cellular and molecular dysfunction at this time point could help identify possible neuroprotection targets.

### 3.2. Altered Burst-Firing Activity of SNc DA Neurons in ParkinQ311X Mice

To test whether parkinQ311X mice at 1 month of age displayed dysfunctional SNc DA neurons, we did an in vivo recording of the spontaneous activity of SNc DA neurons in WT and parkinQ311X mice (*n* = 45 SNc DA neurons from 10 WT mice, *n* = 49 neurons from 10 parkinQ311X mice). SNc DA neurons have two modes of in vivo discharge pattern: either a low frequency (1–5 Hz) single action potential discharge in a pacemaker pattern or transient high-frequency bursting (>15 Hz). The pacemaker pattern occurs in the absence of synaptic input, while the transition into a burst/pause pattern results from the combination of intrinsic ion conductance and inhibitory and excitatory synaptic input [24]. Representative histograms of spontaneous single-neuron firing rate activity are shown in Figure 2a. No significant difference in the tonic firing activity of SNc DA neurons (Figure 2b; *t* = 1.223, df = 93, *p* = 0.224) or in their coefficient of variation (Figure 2c; *t* = 1.216, df = 93, *p* = 0.227) was found between WT and parkinQ311X mice. The percentage of spikes in burst (Figure 2d; *t* = 2.675, df = 61, *p* = 0.0096) and the number of spikes per burst (Figure 2e; *t* = 2.024, df = 61, *p* = 0.0473) were higher in parkinQ311X mice than in WT controls. The mean intraburst frequency (Figure 2f; *t* = 0.9791, df = 61, *p* = 0.331) and the mean burst length (Figure 2g; *t* = 0.6126, df = 61, *p* = 0.542) were similar between genotypes. These patterns point to an early dysregulation of the burst-firing pattern of SNc DA neurons in parkinQ311X mice.

### 3.3. Mitochondrial Dysfunction in SNc DA Neurons of ParkinQ311X Mice

Several lines of evidence show that parkin regulates mitochondrial network quality and is involved in the entire spectrum of the mitochondrial life cycle, including organelle biogenesis, fusion/fission, and clearance via mitophagy [25,26,27]. To test whether the DA neurons of parkinQ311X mice show early mitochondrial dysfunction, we analyzed the OPA1 levels in lysates prepared from the SNc of parkinQ311X mice and WT littermates at 1 month of age (nigra dissection was performed as described previously [9]). The OPA1 protein exists in two different isoforms: the OPA1-long form, tethered to the inner mitochondrial membrane, and the OPA1-short form, soluble in the intermembrane space. Mitochondrial damage such as the loss of mitochondrial potential enhances the proteolytic processing of OPA1, leading to the accumulation of the short isoforms [28]. We found that both the OPA1-long and the OPA1-short isoforms were expressed in SNc and that the OPA1-short/OPA1-long ratio was increased in the SNc of parkinQ311X mice (OPA1short/OPA1long: WT 1.00 ± 0.03 *n* = 6 mice, parkinQ311X 1.20 ± 0.03 *n* = 6 mice, two-tailed unpaired Student’s *t*-test *p* = 0.0019 *t* = 4.164 df = 10, Figure 3a). No differences in the levels of the mitochondrial proteins SDHA and VDAC were found (SDHA: WT 1.00 ± 0.09 *n* = 6 mice, parkinQ311X 0.91 ± 0.03 *n* = 6 mice, two-tailed unpaired Student’s *t*-test *p* > 0.05; VDAC: WT 1.00±0.06 *n* = 6 mice, parkinQ311X 0.91 ± 0.03 *n* = 6 mice, two-tailed unpaired Student’s *t*-test *p* > 0.05, Figure 3a).

This suggests that while the total mitochondrial content was equal in parkinQ311X mice and WT controls, the mitochondria in parkinQ311X DA neurons showed enhanced proteolytic processing of OPA1. Since OPA1 regulates mitochondrial cristae morphology, these data indicate that the expression of the parkinQ311X variant induces pathological changes in mitochondrial morphology. To confirm this hypothesis, we analyzed mitochondrial morphology by transmission electron microscopy (TEM) in isolated SNc tissues from mice at 1 month of age. Mitochondria in control specimens showed the typical rounded/tubular morphology, whereas many mitochondria in parkinQ311X SNc displayed a marked disruption of their ultrastructure. About 26% of the mitochondria in parkinQ311X DA neurons displayed an almost complete alteration of the outer mitochondrial membrane. About 15% of mitochondria in parkinQ311X DA neurons retained their outer membrane integrity but displayed ultrastructural evidence of damage, including the dilation of intracristae spaces, loss of matrix density, and deposits of electron-dense material within the matrix. Based on this constellation of features, these mitochondria can be defined as vesicular mitochondria [29]. Swollen clear mitochondria were also found, showing herniation of the swollen matrix covered by the inner membrane through ruptures in the outer membrane (Figure 3b). We quantified the number of mitochondria with normal morphology, those characterized by loss of the outer membrane, those with vesicular morphology, and the swollen ones. Abnormal mitochondrial morphology was also found in a subset of WT DA neurons. However, the percentage and distribution differed significantly between WT and parkinQ311X DA neurons. The percentage of mitochondria with a normal morphology was 64% in WT mice and 41% in parkinQ311X transgenic. Moreover, 26% of mitochondria lacked an outer membrane in parkinQ311X mice versus 14% in controls; a vesicular morphology was detected in 15% of parkinQ311X mitochondria versus 9% in WT ones; the percentage of swollen mitochondria was 11% in WT and 16% in parkinQ311X mice, respectively (*p* < 0.0001, df 3, chi-square 64.58, Figure 3b). These results revealed the presence of early dysmorphic features in the SNc DA neurons’ mitochondria of 1-month-old parkinQ311X mice.

### 3.4. Cytoplasmic Vacuolization in SNc DA Neurons of ParkinQ311X Mice

Cytoplasmic vacuolization, a morphological phenomenon that often accompanies cell dysfunction and death [30], has been observed in iDA bearing *PARK2* mutations [12]. We observed brain slices from WT and parkinQ311X mice at 1 month of age labeled for TH and found extensive cytoplasmic vacuolization in the SNc DA neurons of parkinQ311X mice (Figure 4a). The average number of vacuoles per cell was: 0.5 (range 0–2) in WT SNc DA neurons and 2.6 (range 0–7) in Q311X SNc DA neurons. The difference was statistically significant (*n* = 51 cells analyzed for each genotype, 3 mice for each genotype, Mann–Whitney test, *p* < 0.0001, Figure 4a). Since in this transgenic mouse model the parkinQ311X variant is expressed in all DA neurons (the transgene is expressed under a DAT promoter) [7], we also analyzed DA neurons in the ventral tegmental area, a brain area that does not undergo degeneration in PD. One to two vacuoles per cell were observed in a small subset of DA neurons of the ventral tegmental area in the tissues of WT and Q311X mice at 1 month of age, but no difference was observed between the two genotypes (Figure 4b). This shows that cytoplasmic vacuolization is a feature of the SNc DA neurons of parkinQ311X mice.

## 4. Discussion

This study shows that the parkinQ311X mouse displays an early degeneration of SNc DA neurons, mirroring the early parkinsonism observed in patients carrying the Q311X mutation in the *PARK2* gene [14]. DA neuron death was preceded by early features of mitochondrial dysfunction, extensive cytoplasmic vacuolization, and dysregulation of spontaneous in vivo firing activity.

Since *PARK2* knock out mice do not show DA neuron loss [3], the question arises as to the mechanism underlying the early degeneration of SNc DA neurons in parkinQ311X mice. Several lines of evidence indicate that knock out mice often develop genetic compensation [31,32], an as yet unexplained phenomenon that may underlie the lack of phenotype despite the absence of a key gene product. An analogous phenomenon occurred in the *PARK2* knock out rat model: these rats displayed normal behavior, no neurochemical changes, and no changes in the number of DA neurons [33]. The molecular mechanisms underlying genetic compensation remain elusive [31]; wired into the network of gene expression control are mechanisms that compensate for gene dosage and that probably undergo activation in knock out mice but not in models expressing physiological levels of WT or mutant proteins. We speculate that *PARK2* knock out mice develop genetic compensation, whereas the transgenic model parkinQ311X does not.

By what mechanism does the expression of the parkinQ311X variants induce DA neuron dysfunction and death? According to the former characterization [7], this mouse expresses comparable levels of endogenous parkin and human parkin variant Q311X, so it is in a genetic condition of heterozygosity. We may hypothesize that this mutant exerts a dominant negative effect similar to that seen in other parkin variants [8,34] and other variants in genes associated with recessive PD forms [35]. Another plausible explanation is that the parkinQ311X mutant exerts a toxic gain of function effect. Both of these possibilities are consistent with the hypothesis that monoallelic *PARK2* mutations increase the risk of PD in humans [36].

Looking for early signs of dysfunction in SNc DA neurons, we found extensive cytoplasmic vacuolization in parkinQ311X mice. Cytoplasmic vacuolization is a morphological phenomenon observed in mammalian cells after exposure to bacterial or viral pathogens, as well as to toxic compounds [30]. Numerous examples of the association between vacuolization and cell death have been reported. However, vacuolization can also be part of a programmed cell response that modulates pro-survival effects [30]. In neurons, vacuolization was found to be associated with ischemic damage [37], treatment with chemical toxins [38], virus infection [39], cholesterol accumulation [40], and neurodegenerative diseases, such as Alzheimer’s disease [41,42]. In DA neurons, vacuolization was found to be associated with methamphetamine neurotoxicity [43]. Neuron vacuolization was also found in the PD brain [44,45] and in cellular models of PD [46]. Although the identity of the vacuoles found in the SNc DA neurons of parkinQ311X mice remains to be clarified, this finding replicates the cytoplasmic vacuolization found in *PARK2* knocked out human DA neurons [12]. The discovery of this cellular anomaly in the brains of animals at 1 month of age indicates that it is a very early dysfunction, but whether this phenomenon is part of a programmed cell death mechanism needs to be clarified. Many cell death pathways are characterized by cytoplasmic vacuolization, including paraptosis, necroptosis, and lysosomal/autophagic-related death mechanisms [47]. Thus, the cytoplasmic vacuolization we observed in parkinQ311X mice might be related to these phenomena.

Other morphological features of dysfunction in SNc DA neurons are mitochondrial abnormalities. Although dysmorphic mitochondria were also found in the SNc neurons of WT mice, the quantitative analysis disclosed a higher percentage of these alterations in parkinQ311X neurons. Since electron microscopy is highly sensitive in detecting even small morphological abnormalities, part of the mitochondria identified as abnormal may actually have had normal functionality.

Parkin is known to play a key role in controlling mitochondrial turnover [25,48,49]; therefore, the accumulation of damaged mitochondria is not surprising. A quantitative ultrastructural analysis of mitochondria previously performed in the brain of *PARK2* knock out mice disclosed damaged mitochondria in mice at 8–12 and 14 months of age [50,51]. However, this is the first evidence of very early damage in a *PARK2* mouse model.

Finally, we noted an anomalous burst-firing activity in the SNc DA neurons of parkinQ311X mice. The in vivo firing activity of adult SNc DA neurons results from the integration of intrinsic pacemaker mechanisms with converging synaptic inputs. Pacemaker activity is self-generated and supported by Ca^2+^ channels. Synaptic inputs (excitatory and inhibitory) generate bursts or transient pauses of electrical activity and are enabled by postsynaptic channels that either dampen or facilitate bursting as key switches [24].

Our previous study on SNc slices revealed an alteration of the pacemaker activity in the SNc DA neurons of parkinQ311X mice [9]. Here, we demonstrate that synaptic-input-dependent activity is likely altered. We hypothesize that such abnormal neuronal excitability results from the dysregulation of postsynaptic channels, e.g., the kainate receptor [9,52]. Since both pacemaker and burst activities are regulated by cytosolic Ca^2+^, and cytosolic Ca^2+^ levels are regulated by mitochondria buffering, it is possible that the abnormal burst-firing activity of the SNc DA neurons in parkinQ311X mice is a consequence of mitochondrial dysfunction. Further studies are needed to support this hypothesis.

Overall, our findings highlight very early abnormalities in the SNc DA neurons of parkinQ311X mice and show that this mouse model is a useful tool to gain insight into the molecular mechanism(s) through which *PARK2* mutations lead to neurodegeneration.

From the genetic point of view, the Parkin Q311X transgenic model does not perfectly reflect ARJP genetics. Nevertheless, no mouse model more faithful to the pathology is currently available, which makes the Parkin Q311X mouse very useful for studying the role of parkin in parkinsonian neurodegeneration. The model holds promise for use in the screening of drugs that may prove capable of rescuing this phenomenon. A future area of focus is to create new mouse models that express other parkin mutations as a transgene or in endogenous DNA and to determine whether they are associated with a neurodegenerative phenotype.

## Figures and Tables

**Figure 1 biomedicines-09-00514-f001:**
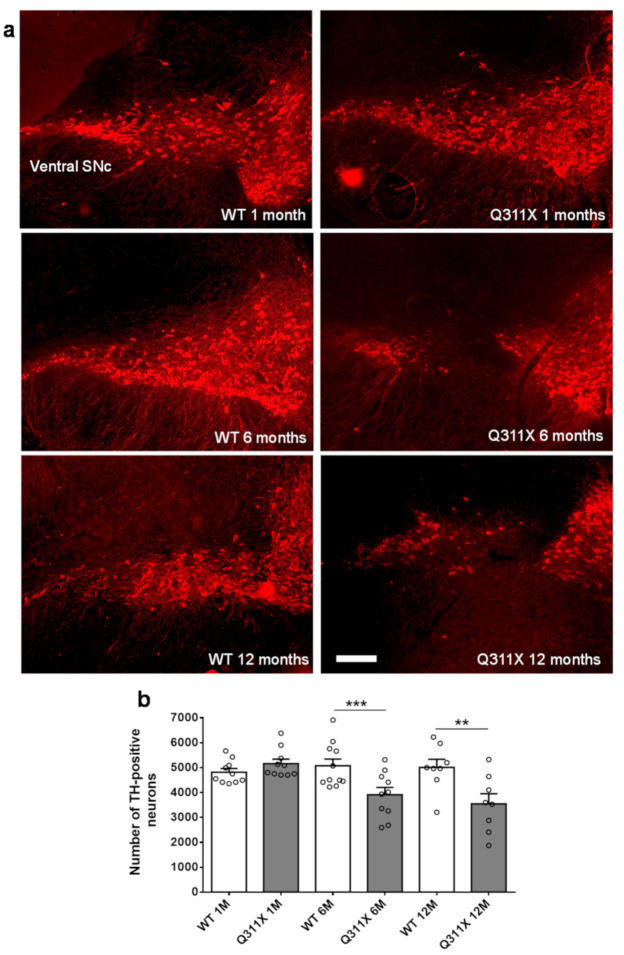
ParkinQ311X mice display early SNc DA neuronal loss. (**a**) Representative immunofluorescence images showing tyrosine hydroxylase (TH) labeling in the SNc of WT and parkinQ311X mice at 1, 6, and 12 months of age. At 6 and 12 months of age, a loss of DA neurons and dopaminergic dendrites in the pars reticulata of the substantia nigra becomes apparent. (**b**) The graph presents a stereological count of DA neurons. Data are mean ± SEM of *n* mice: WT 1 month (1M) 4817 ± 146 *n* = 10, parkinQ311X mice (1M) 5160 ± 182 *n* = 10, WT 6 months (6M) 5075 ± 270 *n* = 11, parkinQ311X mice (6M) 3913 ± 293 *n* = 10, WT 12 months (12M) 5009 ± 326 *n* = 8, parkinQ311X mice (12M) 3546 ± 406 *n* = 8; one-way ANOVA followed by Sidak’s multiple comparisons test WT 6M vs. ParkinQ311X 6M *** *p* < 0.001, WT 12M vs. ParkinQ311X 12M ** *p* < 0.01. Bar is 100 μM.

**Figure 2 biomedicines-09-00514-f002:**
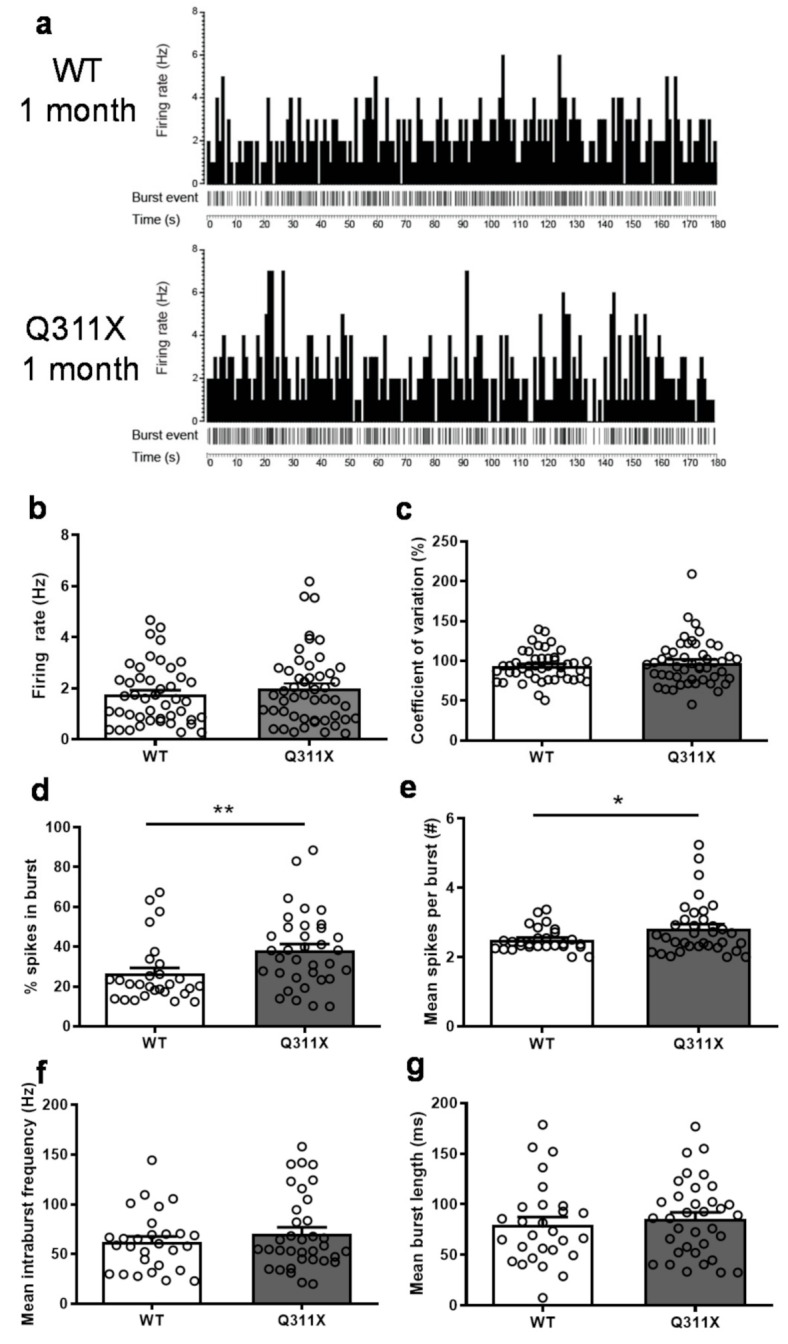
Burst activity of SNc DA neurons is increased in parkinQ311X mice. (**a**) Representative histograms of spontaneous single-neuron firing rate activity (first trace) and burst event (second trace) in WT and parkinQ311X mice at 1 month of age. (**b**,**c**) The mean spontaneous firing activity and the coefficient of variation (COV) of SNc DA neurons were similar in WT and parkinQ311X mice. (**d**,**e**) The percentage of spikes in burst and the number of spikes per burst were higher in parkinQ311X mice than in WT mice. (**f**,**g**) The mean intraburst frequency and the mean burst length were similar in WT and parkinQ311X mice. Data are mean ±SEM of 45 neurons from 10 WT mice and 49 neurons from 10 parkinQ311X mice. * *p* < 0.05 and ** *p* < 0.01, different from WT mice (unpaired Student’s *t*-test).

**Figure 3 biomedicines-09-00514-f003:**
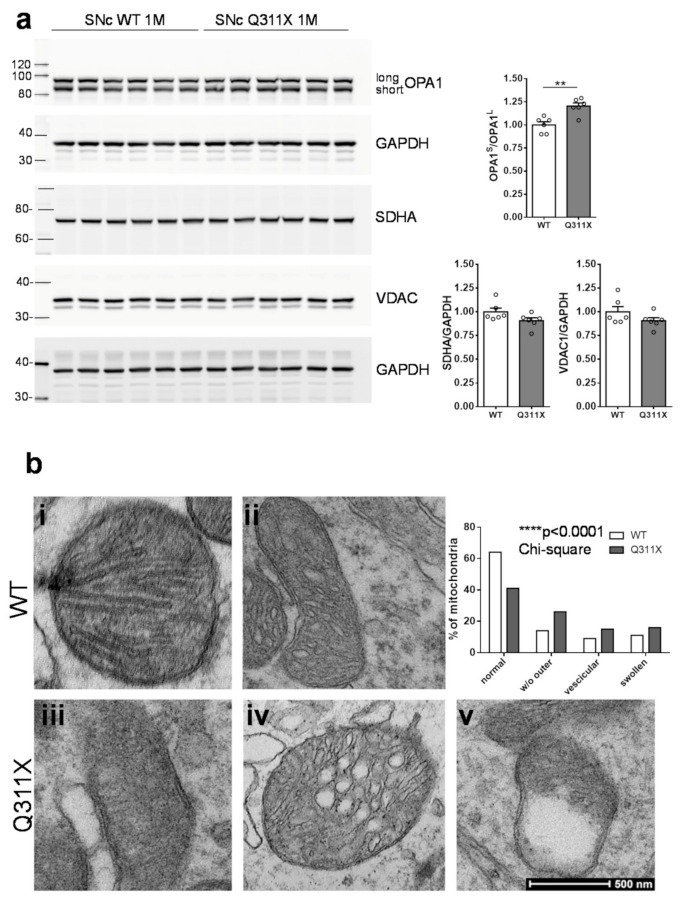
Expression of the human parkin variant parkinQ311X induces early mitochondrial dysfunction in SNc DA neurons. (**a**) Representative Western blots showing the levels of mitochondrial proteins (OPA1, SDHA, VDAC) in lysates prepared from the substantia nigra of WT or parkinQ311X mice at 1 month of age. Samples were run in triplicate, with each lane loaded with lysate from an individual mouse. The histograms on the right show the mean ±SEM calculated from densitometer quantification (** *p* < 0.01 unpaired Student’s *t*-test). (**b**) Representative TEM images showing mitochondria in SNc DA neurons of WT (control, **i,ii**) and parkinQ311X (**iii,iv,v**) mice. The mitochondria in control specimens show the typical rounded/tubular morphology (**i,ii**). The mitochondria in parkinQ311X mice display marked ultrastructure disruption (**iii,iv,v**) and a lack of the typical crystal organization found in the control tissue: 26% of the mitochondria in parkinQ311X DA neurons displayed an almost complete loss of outer membrane shape and form (**iii**). Other mitochondria in the parkinQ311X DA neurons retained outer membrane integrity but demonstrated ultrastructural damage: dilation of intracrystal spaces, loss of matrix density, and deposits of electron-dense material within the matrix (vesicular mitochondria) (**iv**). Swollen, clear mitochondria showed herniation of the swollen matrix covered by the inner membrane through the ruptured regions of the outer membrane (**v**). The histograms on the right show the distribution of morphological features. For each section we counted the number of mitochondria with normal morphology, those with loss of outer membrane, those with vesicular morphology, and the swollen ones. WT and parkinQ311X mice (*n* = 3 in each group) were analyzed: total count *n* = 246 for WT and *n* = 320 mitochondria for parkinQ311X mice. There was a statistically significant difference in the distribution of WT and parkinQ311X neurons (**** *p* < 0.0001, df 3, chi-square 64.58).

**Figure 4 biomedicines-09-00514-f004:**
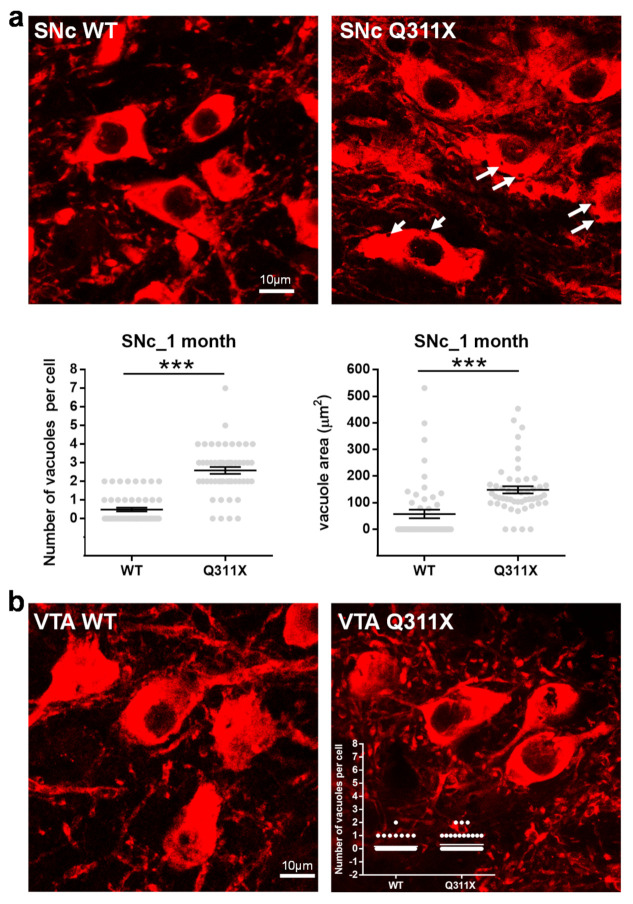
SNc DA neurons in parkinQ311X mice display cytoplasmic vacuolization. (**a**) Representative images showing DA neurons (TH labelling) in the SNc of WT and parkinQ311X mice at 1 month of age. The white arrows indicate the vacuoles in the cytoplasm of DA neurons. The graphs below the images show the number of vacuoles and vacuole area. Data are the mean ± SEM of the number of vacuoles per cell and are derived from *n* = 3 mice per genotype: WT 1M SNc 0.49 ± 0.11, parkinQ311X 1M SNc 2.59 ± 0.19 (*n* = 51 cells analyzed for each genotype, Mann–Whitney U test, *** *p* < 0.001). (**b**) Representative images showing DA neurons of the ventral tegmental area. One to two vacuoles per cell were observed in a small subset of DA neurons of the ventral tegmental area in WT and parkinQ311X mice, but no difference between the two genotypes was observed.

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
