# Peer review of "Early Dysfunction of Substantia Nigra Dopamine Neurons in the ParkinQ311X Mouse"

_biomedicines, 2021, doi:10.3390/biomedicines9050514_

Round 1

Reviewer 1 Report

Well written simple characterization paper

comments:

What are the levels of Park protein in wt and mutant mice? is there comparable levels of expression? if not does this contribute to effects seen?

Localization of the mutant and WT Park? Possible miss localization as a contributing factor?

Is there an increase, decrease or mislocalization of the mutant park in dystrophic mitochondria?

Figure 3 image seems messed up with a blot missing

How many animals were used for counting the mitochondria? Just shown is a total not average percentage per mouse

Reviewer 2 Report

Review for Biomedicines-1188179

This paper characterized the early changes in dopaminergic neurons (DNs) in Substantia Nigra pars compacta (SNc) in a parkin-based genetic model.  The model was developed by the Yang group in 2009 (Ref #7) and since intensively studied around the world.

The model involves BAC transgenetic mice expressing human mutant ParkinQ311X, which develops both age-dependent DNs phenotypes and slowly progressive motor phenotypes that start at ~16-month of age.  This research was focused on the DNs changes at 1-month old that precede neurodegeneration.

It reported DNs changes in three aspects:

  1. Spontaneous burst-firing activities, measured by in vivo single-unit extracellular recordings.
  2. Mitochondrial morphology, observed through transmission electron microscopy.
  3. Number of vacuoles in TH+ neurons in SNc, using Immuno image analysis.

The experiments were well performed and thoroughly analyzed. It will provide valuable information for the study of the dominant toxic mechanism elicited by the ParkinQ331X mutant.

However, the ParkinQ311X model is not an early onset model: it does not demonstrate behavioral changes until 16-month old while juvenile Parkinsonism patients have the onset at ~20 years old.  Early dysfunction in this model might not be relevant to the ARJP and the paper did not compare the results with early DN changes in other mouse models based on known PD mutations (except for PARK2-KO).

Specific points:

  1. Line 29-30, need references for which early key features of ARJP were recapitulated by the results in this paper.
  2. Line 75, there was a reported (PMID: 10984671) sex difference in ARJP, in which M < F.
  3. Figure 1, Reference #7 reported no TH loss at 3 months of age. Need discussion on why chose 1-month instead of 3-month old mice.
  4. Line 238, loss of TH immunoactivity at 6 months is not equal to the loss of DA neurons, although turning-off TH is a sign of DA degeneration.
  5. Figure 2, no tonic-firing difference is contradicting to findings in Reference #9, which need more discussion.
  6. Figure 3, the Chi-square result needs to be labeled in the figure for better understanding.
  7. Figure 3, needs an explanation why ~40% of mitochondria observed in the WT mice at 1 month of age were abnormal.
  8. Figure 4, that cytoplasmic vacuolization is an important sign. However, when DNs are degenerating, they will turn off TH expression early on, making vacuolization comparison at different stages difficult. Need more reference on tracking vacuolization in PD progression.
  9. Line 441, need references on the association between vacuolization and cell death, especially in dopamine neurons.

Author Response

Ref2

We thank referee#2 for considering our manuscript and for giving us the opportunity to improve it

This paper characterized the early changes in dopaminergic neurons (DNs) in Substantia Nigra pars compacta (SNc) in a parkin-based genetic model.  The model was developed by the Yang group in 2009 (Ref #7) and since intensively studied around the world.

The model involves BAC transgenetic mice expressing human mutant ParkinQ311X, which develops both age-dependent DNs phenotypes and slowly progressive motor phenotypes that start at ~16-month of age.  This research was focused on the DNs changes at 1-month old that precede neurodegeneration.

It reported DNs changes in three aspects:

  1. Spontaneous burst-firing activities, measured by in vivo single-unit extracellular recordings.
  2. Mitochondrial morphology, observed through transmission electron microscopy.
  3. Number of vacuoles in TH+ neurons in SNc, using Immuno image analysis.

The experiments were well performed and thoroughly analyzed. It will provide valuable information for the study of the dominant toxic mechanism elicited by the ParkinQ331X mutant.

However, the ParkinQ311X model is not an early onset model: it does not demonstrate behavioral changes until 16-month old while juvenile Parkinsonism patients have the onset at ~20 years old.  Early dysfunction in this model might not be relevant to the ARJP and the paper did not compare the results with early DN changes in other mouse models based on known PD mutations (except for PARK2-KO).

As requested we modified the discussion, page 22 lines 459-462 and wrote that “From the genetic point of view the Parkin Q311X transgenic model does not perfectly reflect ARJP genetics. Nevertheless, no mouse model more faithful to the pathology is currently available, which makes the Parkin Q311X mouse very useful for studying the role of parkin in parkinsonian neurodegeneration”. As requested, to better compare our data with previous literature, we also added in the discussion information about the rat parkin KO model page 20 lines 394-396. Other available PARKIN models were not discussed because they were non mammalian models of drosophila (PMID: 12642658; PMID: 15911761; PMID: 15073152; PMID: 17687034) and c. elegans (PMID: 16239214).

Specific points:

  1. Line 29-30, need references for which early key features of ARJP were recapitulated by the results in this paper. We did not have the possibility to add references at line 29-30 because references are not allowed in the abstract section, however we added a reference (“The neuropathology of genetic Parkinson's disease”, PMID: 22451330) page, line 246.
  1. Line 75, there was a reported (PMID: 10984671) sex difference in ARJP, in which M < F.  This paper, published in 2000, described ARJP cases from only 13 families and they found M£F. However, the PARKIN gene is localized on autosome chromosome 4, the mutations are equally transmitted to male and female, the penetrance is 100% for pathological variants (all the identified mutations can be found in the following database https://www.mdsgene.org/d/1/g/4)
  2. Figure 1, Reference #7 reported no TH loss at 3 months of age. Need discussion on why chose 1-month instead of 3-month old mice. In humans, the disease due to mutations in the PARKIN gene is associated with pathology with juvenile onset. The median age at onset is 31 years, however in many cases the symptoms present before 20 years of age and at this age there is already neurodegeneration (Brüggemann N, Klein C. Parkin Type of Early-Onset Parkinson Disease. 2001 Apr 17 [Updated 2020 Apr 23]. In: Adam MP, Ardinger HH, Pagon RA, et al., editors. GeneReviews® [Internet]. Seattle (WA): University of Washington, Seattle; 1993-2021. Available from: https://www.ncbi.nlm.nih.gov/books/NBK1478/). We looked in the literature for age relation between mice and humans and found that 20 human years can be equated to about 65-70 days (Men and mice: Relating their ages. PMID: 26596563). For this reason we decided to analyse the mouse brain tissues at 30 days, in order to highlight alterations that precede the neurodegenerative phenomenon. Furthermore, the stereological counting data that we have indicate that neurodegeneration exists at 6 months and not at 1. We did not perform the counts also at 3 months in order not to dramatically increase the number of animals used in the study. Since we were certain that at 1 month of age there was no neurodegeneration, we have performed all the experiments at this age.
  3. Line 238, loss of TH immunoactivity at 6 months is not equal to the loss of DA neurons, although turning-off TH is a sign of DA degeneration. As requested, we modified the sentence, instead of “loss of DA neurons”, we wrote “degeneration of DA” as suggested, page lines 244-245.
  4. Figure 2, no tonic-firing difference is contradicting to findings in Reference #9, which need more discussion. We thank the Reviewer for this observation. Our previous study (Reference #9, Regoni et al., 2020) analysed DA neurons in deafferented mesencephalic slices in vitro. The present study analyses the neuronal firing in vivo, i.e. in an intact system of a living animal (under anaesthesia). The different models might explain why no differences in tonic firing can be seen in vivo. In fact, DA SNc neurons receive glutamatergic and GABAergic inputs from many brain regions. We can speculate that the difference between WT and parkinQ311X in the pacemaker rate is observable only in deafferent slices because the integration of intrinsic pacemaker mechanisms with converging synaptic inputs may not allow to see the difference.
  5. Figure 3, the Chi-square result needs to be labeled in the figure for better understanding. As requested, we indicated in figure 3 that data were analysed by Chi-square and the p value.
  6. Figure 3, needs an explanation why ~40% of mitochondria observed in the WT mice at 1 month of age were abnormal. Electron microscopy is sensitive in detecting even small morphological abnormalities. It is possible that part of the mitochondria identified as morphologically abnormal actually has normal functionality. However, we have chosen to use electron microscopy precisely to identify very early alterations in parkinQ311X tissues. As requested, we have discussed this issue in discussion (page 21, lines 433-435).
  7. Figure 4, that cytoplasmic vacuolization is an important sign. However, when DNs are degenerating, they will turn off TH expression early on, making vacuolization comparison at different stages difficult. Need more reference on tracking vacuolization in PD progression. As requested we searched in the literature other studies showing vacuolization in PD or in DA neurons and we added new references (44-47) in the discussion. We identified DA neurons by TH labelling because it is the best-validated method for stereological analyses. However, we agree with the Reviewer that changes of TH levels can occur in DA neurons. Indeed, we have been investigating this issue in another project. We thank the Reviewer for this comment.
  8. Line 441, need references on the association between vacuolization and cell death, especially in dopamine neurons. As requested, we searched in the literature other studies showing vacuolization in PD or in DA neurons. The following two new sentences were added in Discussion (page 21, lines 419-421): “In DA neurons, vacuolization was found to be associated with methamphetamine neurotoxicity [44]. Neuron vacuolization was also found in PD brain [45,46] and in in vitro models of PD [47]”.

Round 2

Reviewer 1 Report

Answered my concerns with the paper adequately.